# Poly(2-Hydroxyethyl methacrylate-co-N,N-dimethylacrylamide)-Coated Quartz Crystal Microbalance Sensor: Membrane Characterization and Proof of Concept

**DOI:** 10.3390/gels7040151

**Published:** 2021-09-24

**Authors:** Angel Ramon Hernandez-Martinez

**Affiliations:** Centro de Física Aplicada y Tecnología Avanzada, Universidad Nacional Autónoma de México (UNAM), Boulevard Juriquilla 3001, Queretaro 76230, Queretaro, Mexico; angel.ramon.hernadez@gmail.com or arhm@fata.unam.mx

**Keywords:** hydrogels, copolymers, HEMA, DMAa, membranes, QCM-sensors

## Abstract

Application-oriented hydrogel properties can be obtained by modifying the synthesis conditions of the materials. The purpose of this study is to achieve customized properties for sensing applications of hydrogel membranes based on poly(2-hydroxyethyl methacrylate), HEMA and N,N-dimethylacrylamide, DMAa. Copolymer p(HEMA-co-DMAa) hydrogels were prepared by varying the DMAa monomer ratio from 0–100% in 20% increments. Hydrogel membranes were characterized by attenuated infrared spectroscopy. Swelling and sorption were evaluated using cation solutions. Copolymers were also synthesized on the gold surface of quartz crystal microbalances (QCM) as coating membranes. A proof of concept was conducted for approaching the design and development of QCM sensors based on P(DMAa-co-HEMA)-membranes. Results showed that the water and ion adsorption capacity of hydrogel membranes increased with higher DMAa content. Membranes are not selective to a specific location but did show different transport features with each cation. The QCM coated with the selected membrane presented linear relationships between resonance frequency and ions concentration in solution (10–120 ppm). As a consequence, hydrogel membranes obtained are promising for the development of future biosensing devices.

## 1. Introduction

Diagnostic testing provides critical insight to support decisions regarding treatment and referral to secondary care. In this sense, the development of sensing platforms for diagnosis such as chemical sensors with high sensitivity, precision, selectivity, reproducibility, and low false positives, is a technological need that must be fulfilled. Biosensors are chemical sensors that provide direct information about the composition of their environment; and consist of a selective membrane or layer associated with a physical transducer [1]. This selective membrane involves some type of biological material as a biological recognition element (biochemical receptor) immobilized on or within it, providing sensitivity and selectivity [2,3,4].

Therefore, the selective membrane is a key component in biosensors as the recognition element and the transducer. Consequently, the membranes play a critical role in the design and development of chemical sensors. As components of biosensors, membranes have more than one function, thus they have been used extensively for; (i) immobilizing biomolecules and/or cofactors as ionophores, (ii) improving selectivity of the chemical sensor, (iii) controlling the enzyme kinetics, (iv) improving biocompatibility of the system, and (v) controlling electron transfer properties of the system [4,5,6,7,8].

In this context, polymeric membranes have been used in the design of biosensors and they can even act as the sensing element in chemical sensors [9,10]. Polymeric membranes formed by hydrogels are capable of adsorbing and retaining a considerable amount of water or biological fluids [11]; thus, they have been used in various forms and applications, such as scaffolds [12,13], injectable materials [14], microgels [15], nanoparticles or nanofibers [16,17], and as membranes. Hydrogel membranes (HMs) are constructed by hydrophilic polymers for a broad range of applications, including its use and assembly in sensing devices [18]. Some studies explored HMs as immobilization or sensor membranes, or as materials to improve the selectivity of the chemical sensor due to their physicochemical properties, such as their water-swollen features, biocompatibility, and special transport phenomena [18,19,20,21,22,23,24,25,26,27,28,29]. Copolymers based on 2-hydroxyethyl methacrylate (HEMA) have been used as HMs; examples of them are: poly(2-hydroxyethyl methacrylate-co-N-methacryloyl-(L)-phenylalanine methyl ester), i.e., p(HEMA-co-MAP) for its use on albumin depletion studies [30]; poly(2-hydroxyethyl methacrylate-co-ethylene dimethacrylate), i.e., poly(HEMA-EDMA) for immunoglobulin G (IgG) [31]; and a composite of PHEMA with polypyrrole (PPy) used to develop a cholesterol biosensor [29]. PHEMA has become the main component of various hydrogels and copolymers (for several applications), because of its excellent biocompatibility, and has been used in various applications such as drug delivery, soft contact lenses manufacturing, and materials for female breasts and nasal cartilages reconstruction, artificial corneas, wound dressings, and porous sponges that promote cellular ingrowth [32,33,34,35,36,37,38,39].

Reported advantages about PHEMA-based improved materials motivate my research group to develop novel materials based on 2-hydroxyethyl methacrylate (HEMA) for various applications. Consequently, the copolymerization of PHEMA with other nonionic acrylic monomers such as the N, N-dimethylacrylamide (DMAa) has been explored. DMAa is an interesting monomer; it is the simplest among the N,N-disubstituted self-crosslinking acrylamides, used in biological applications due to its superadsorbent properties (e.g., elongation up to 1350%) [40], with multiple applications [41,42,43]. As previously reported, poly(N,N-dimethylacrylamide-co-2-hydroxyethyl methacrylate) [P(DMAa-co-HEMA)], with a monomeric ratio of 1:1 mol%, was synthesized and studied; and it resulted in being an efficient hydrogel for adsorbing and removing methylene blue [44] and lead (II) [45] from aqueous solutions. Based on these previous results, it was proposed to explore the modification of the physical properties and fit the HEMA-based materials to an end-use application through the copolymerization of HEMA with DMAa. In that sense, this report is our first approach to exploring the hypothesis that establishes that the transport properties such as swelling and adsorption of PHEMA-based membranes can be modified through copolymerization using DMAa at different molar ratios to achieve custom-made properties for sensing and biosensing applications. Therefore, the aim of this work is to synthesize a series of P(DMAa-co-HEMA) hydrogel membranes, at different molar ratios of the comonomers, to gain insight into the effects of these constituting elements (monomers) on the overall swelling and sorption properties, as well as in their transport phenomena. Finally, a proof of concept was established to explore and illustrate the way in which a working quartz crystal microbalance-type (QCM-type) sensor based on these P(DMAa-co-HEMA)-membranes may be designed and fabricated.

Therefore, an initial attempt to obtain a Poly(2-hydroxyethyl methacrylate-co-N,N-dimethylacrylamide)-coated quartz crystal microbalance (QCM) sensor is reported.

## 2. Results and Discussion

### 2.1. Synthesis of the Copolymer Membranes and Chemical Characterization

Polymerization was carried out through a free radical mechanism, whereby AIBN decomposes into two free radicals that rapidly react with HEMA or DMAa vinyl groups; therefore, polymerization can be confirmed using spectroscopic techniques such as ATR-FTIR [36,37,44,45]. Figure 1 shows the IR spectra of each copolymer; from the sample with 100%mol of HEMA to the sample with 100%mol DMAa. Adsorption bands related to the vinyl C=C groups (~3100 cm^−1^ and 3020 cm^−1^) of monomers of dimethylacrylamide (DMAa) and 2-hydroxyethyl methacrylate (HEMA) are absent in the final copolymer. On the other hand, bands associated with the normal modes of vibration of C-H bonds (~2900 cm^−1^; νCH3 and δCH3) are shown in all spectra. Broadbands extending from ~3100 cm^−1^ to 3500 cm^−1^ are associated with –OH stretching vibrations; the broad absorption band centered at ~3380 cm^−1^ found in all copolymers is related to the –OH functional group inherited from the chemical structure of the HEMA monomer. Meanwhile, the band centered at ~3450 cm^−1^ present on the homopolymer PDMAa spectrum comes from –OH’s of the water molecules adsorbed by the hydrogel as moisture [38,40]. C=O deformation vibrations were found at ~1720 cm^−1^ or ~1610 cm^−1^ in the P(HEMA) and P(DMAa) spectra, respectively (Figure 1). Electronic anisotropy of the surroundings around the carbonyls within each polymer explains the differences in absorptions wavenumber [41]. Unlike the PHEMA and PDMAa homopolymers, the five PHEMA/PDMAa copolymers present two absorption bands (one at ~1720 cm^−1^ and other at ~1610 cm^−1^) associated with the deformation vibrations of C=O. The coincidence in the wavenumber of each of these bands could be an indication that the copolymerization was carried out preferably by blocks of PHEMA and PDMAa, conserving both structural environments around each one of the C=O groups. On the other hand, the variation of intensity in each band seems to be correlated with the concentration of each co-monomer.

### 2.2. Swelling Behavior

Water absorption behavior and limit swelling were studied to determine the physicochemical equilibrium conditions of each HM, and there is evidence that the transport properties such as swelling and adsorption can be modified through copolymerization using DMAa at different molar ratios, as established in the hypothesis and in the main objective. Subsequently, a study was conducted on their transport properties to various analytes (in solution) at the conditions of physicochemical equilibrium.

The profiles of swelling at 25 °C and pH 7 of the HMs are shown in Appendix A. As previously reported [44,45], swelling capacity is affected by temperature and pH, therefore, in this study, all the physicochemical analyses were conducted in deionized water at a constant temperature and constant pH conditions to assess only the effect of the %mol of DMAa content in each HM. From the swelling profiles (Appendix A), the maximum absorption capacity known as swelling limit is obtained, which is achieved when the surface of the polymeric chains of the hydrogel has been completely covered by water molecules. Under these conditions, the hydrogel can no longer adsorb water molecules and the sorption/desorption processes are in physicochemical equilibrium, hence there is no longer any increase in swelling over time. The limit swellings for each membrane as a function of the %mol DMAa content are shown in Figure 2a, while the time required to reach this equilibrium is shown in Figure 2b.

Each point in Figure 2a,b represents the equilibrium conditions reached during the adsorption of water molecules by each membrane. They were obtained from the water adsorption (swelling) profiles of each membrane, which can be reviewed in detail in Appendix A. The physicochemical equilibrium parameters of the membranes, under immersion conditions at 25 °C and pH = 7, are the swelling ratios (Sr) and the immersion time (Te; equilibration time) required to achieve each Sr. The swelling ratios as a percentage are shown in Figure 2a, while the Te are shown in Figure 2b. The effect of the DMAa chains on the water adsorption capacity is evident; the swelling capacities of PHEMA and PDMAa are represented by the limiting values in Figure 2a; approximately 100% for PHEMA and 1400% for DMAa; and as the %mol content of DMAa increases in the HMs, the water adsorption capacity increases with respect to PHEMA. The reason for the enhancement of the swelling capacity at higher molar ratios of DMAa can be due to the incorporation of non-ionic DMAa chains into the hydrogel network, as hydrophilic N-C=O groups increase hydrophilicity of the synthesized hydrogel.

In the case of Te, the limiting values also correspond to the homopolymers PHEMA and PDMAa, but in the case of the copolymers, no increase in Te values is evident with higher DMAa content. With the exception of PDMAa, which required about 35 h to reach maximum swelling, the rest of the membranes only required between 10 and 15 h to reach equilibrium and maximum swelling. These results are coherent for the case of block copolymers; because the higher the DMAa content, the longer the PDMAa chains will be, and this explains the increase in swelling capacity. On the other hand, the time needed to achieve physicochemical equilibrium will be the same regardless of the amount of DMAa blocks present in polymer chains. Previous reports [44,45] have shown nuclear magnetic resonance data pointing to a predominantly block copolymerization between DMAa and HEMA, which is congruent with the results in Figure 2a,b.

### 2.3. Sorption Study

The effect of contact time on Pb(II) adsorption was initially evaluated, as it is commonly performed in studies of adsorption processes of chemical species (in this case, cations). This is important because it must be ensured that the equilibrium point was reached in order to report the parameters; Qe and Te, where the former represents the mass (in mg) of maximum adsorbed ions (at physicochemical equilibrium) per unit mass (1 g) of membrane, while the latter represents the time needed to reach Qe.

As previously reported, the adsorbed mass of Pb(II) by the hydrogel P(HEMA-co-DMAa; 50 mol%) depends strongly on the initial concentration of metal ions [45]. As an example, Appendix A shows the mass of Pb(II) ions adsorbed per unit mass of the membrane p(HEMA-co-DMAa) with 50% molar DMAa as a function of time (Qt) at 25 °C and pH 8, for an initial concentration of 40 ppm; in this case, the physicochemical equilibrium is reached with a Qe = 31 mg/g at a Te of 3.5 h. Meanwhile, a Qe = 59 mg/g is achieved at 4 h of immersion of the same membrane, but with an initial concentration of 80 ppm in the lead(II) ion solution (Appendix A). Clearly, there is a significant effect on ion adsorption due to the initial concentration. Therefore, it was necessary to evaluate Qe as a function of initial concentration (in all cases, at constant temperature and pH), and Appendix A shows Qe as a function of initial concentration (Ci), within the range of 10–300 ppm Pb(II) ions, for the same P(HEMA-co-DMAa; 50% mol) membrane. In Appendix A, it can be seen that Qe is higher at higher Ci, reaching a plateau at about Qe,max = 65 mg/g, with a Co of 120 ppm Pb(II).

Similarly, it was necessary to evaluate whether this performance of the membrane with 50 mol% DMAa was repeated for the other membranes. Appendix A shows similar behavior for the 80 mol% DMAa membrane; however, for this membrane, the maximum Qe (Qe,max) was about 90 mg/g with a Ci of 180 ppm Pb(II). Therefore, each of the membranes, from 0 mol% to 100 mol% DMAa, had to be tested to obtain its adsorption profile as a function of the initial concentration (Qe vs. Ci) with each of the five cations. All the information obtained from this extensive adsorption study is shown condensed in the images in Figure 3 and with more detail in Appendix A.

### 2.4. Sorption Equilibrium

As described above, as the initial ion concentration was gradually increasing from 10 to 300 ppm, the amount of ions adsorbed at equilibrium increased, i.e., the equilibrium shifted to higher values with higher Ci. As a consequence, different values of Qe from the adsorption isotherms were obtained when solutions of higher initial concentration (Ci) were used, until a critical point was reached, at which a significant change in Qe was not detectable (see the adsorption isotherms in Appendix A). This critical point represents a Qe,max for each membrane, and it is different for each cation (Figure 3a). Figure 3a shows Qe,max as a function of mol% DMAa content for the six cations; starting, as a lower limit, with the case of PHEMA, and as an upper limit, with the case of PDMAa. Adsorption is enabled by the presence of unbalanced or residual forces on the surface of the polymer chains that attract and retain the molecules of the material in contact. Therefore, the chemical nature and surface area offered by the membranes become important factors to consider. In HMs, the effect of DMAa in the copolymer was evident, i.e., with a higher DMAa content in the membrane, a higher ion adsorption capacity per mass unit of adsorbent was achieved. This can be explained by the increase of hydrophilic functional groups that is propitiated by increasing the DMAa content; the C=O and R–N–(Me)_2_ groups present in DMAa increase the capacity of the membranes to trap cations through intermolecular attractive forces involved in the adsorption mechanism. Therefore, it is a chemisorption compartment; then, the greater the amount of monomeric units with hydrophilic groups (donors), the greater the capacity to trap cations (acceptors). This is consistent for the six cations studied (Figure 3a), however, the Pb(II) cation presented the highest affinity to be adsorbed on the membranes, followed by Ca(II) and Cr(III), while Co(II) was the cation with the lowest Qe. Analyzing the 2^+^ charge cations, the adsorption capacity Qe of the membranes decreases according to the series, Pb(II) > Cd(II) > Cu(II) > Co(II), which can be related to the decrease of the ionic radius which decreases according to, Pb(II)(1.2 A) > Cd(II)(0.97 A) > Co(II)(0.74 A) > Cu(II)(0.69 A) [46,47]. For the case of the 3^+^ charged cations; the Cr(III) (0.69A ionic radius) showed higher Qe than Fe(III) (0.64 A ionic radius). Without being conclusive, there is an apparent relationship between the possible size of the cation and Qe; this could imply that cations with ionic radius approaching to 1A have greater susceptibility to generate interactions with the hydrophilic functional groups present in the polymeric chains of the HEMA/DMAa membranes; apparently, the size of the cation is determinant, as has been argued in previous reports [48,49,50,51].

Figure 3b shows the ratio Qe/Ci versus mol% DMAa contained in each membrane. Qe/Ci is representative of the adsorption efficiency, and Figure 3b shows that DMAa contributes to improving efficiency in HEMA/DMAa membranes. Analysis suggests that cation size has more influence than its charge, considering that Pb(II) has the highest efficiency (Qe/Ci), followed by Cd(II) and Fe(III), depending on which section is analyzed in Figure 3b. For DMAa content lower than 50 mol%, Fe(III) presents higher Qe/Ci ratio, while for concentrations higher than 50 mol%, Cd(II) has a better efficiency.

From the data in Figure 3a,b, it can be concluded that concentrations above 60 mol% have negligible changes in the amount of adsorbed mass of ions per mass unit of adsorbent HMs (Qe), and although the mechanical properties of the membranes were outside the scope of this report, higher amounts of DMAa can significantly decrease the mechanical properties.

On the other hand, the time required to achieve each Qe,max is presented in Figure 3c,d. Results were presented separately for clarity (some sections overlapped). The presence of DMAa units in the membranes had a less evident effect on equilibrium time than on adsorption capacity. Equilibrium time for PHEMA and PDMAa homopolymers is at the end of each line (Figure 3c,d). PHEMA reaches equilibrium with any cation from 3 to 5 h, while for PDMAa, it took from 7 to 14 h of immersion to achieve Qe,max, this is a clear difference. However, for copolymeric membranes, from 20 to 80 mol% DMAa, the equilibrium time was very similar for the same cation; for example, Pb(II) equilibrium time was ca. 6 h for any membrane in that range. The same occurred for Cu(II) ca. 8 h, Co(II) ca. 9 h and Cd(II) ca. 4.5 h (the latter cation is a reference point in Figure 3c,d). The Fe (III) cation is the only one that presents a quasi-direct variation between the equilibrium time and the DMAa content. Cd (II) and Pb (II) cations had the lowest equilibrium time, and achieved the highest Qe and Qe/Ci. While Co (II) and Cu (II) had the largest equilibrium time, and the lowest Qe and Qe/Ci, the Cd (II) and Pb (II) cations have the largest ionic radius, then they should diffuse or be transported between the polymer chains of the membranes with greater difficulty; while the cationic species Co (II) and Cu (II) represent the opposite case. Therefore, the most logical explanation for the shorter equilibrium times corresponding to the larger cations is that they are more easily trapped by the functional groups present in the membranes; while smaller cations are more easily transported between polymeric networks, and due to their smaller volume, they manage to escape between polymeric chains because the molecular interactions are less intense, due to a greater free volume between polymer chains and ions, which leads to a lower attractive force.

The adsorption results of the six cations shown in Figure 3 confirmed that the HMs based on the copolymers of HEMA and DMAa exhibited different transport properties. In this report, the type of kinetics and its association with the various classical semi-empirical models were not explored, because it was not part of the research objective at this stage. It was sought to demonstrate that HMs based on HEMA and DMAa copolymers have potential applications as membranes for the development of sensors and biosensors; either as selective and separation membranes or as membranes to immobilize recognition probes. In this sense, the equilibrium times shown in Figure 3c,d demonstrate differentiated transport processes for the six cations studied; and in conjunction with the different Qe/Ci ratios of Figure 3b, as well as Figure 3a, it is demonstrated that HEMA-co-DMAa HMs have suitable properties to be integrated into sensing devices. The HMs with 40 mol%, 50 mol%, and 60 mol% presented different Qe, max and Qe/Ci for each of the cations studied, as well as differentiated equilibrium times in most cases, each of these three HMs presented only one overlap in the equilibrium time. On the other hand, the HMs with 20 mol% and 80 mol% presented two overlapping equilibrium times. In the case of homopolymers, PHEMA has poor adsorption properties and PDMAa has poor mechanical properties.

### 2.5. Evaluation of QCM-Sensors Coated

A coated QCM type sensor was manufactured using each HM to evaluate its potential application as hydrogel membranes (HMs) for sensors or biosensors, and then one membrane was selected to carry out a proof of concept that validated or rejected the hypothesis proposed for this research. HMs from P(DMAa-co-HEMA) are typical hydrogels, hence their mechanical properties change as they adsorb the aqueous sample for analysis. Each HM achieves its highest elastic consistency at the physicochemical equilibrium point, therefore HMs coating the surface of a QCM will evolve into a softer membrane as the test HM adsorbs an aqueous sample into it. Therefore, the HM coating the gold surface of the QCM will induce greater energy losses as it adsorbs the aqueous sample during its analysis. On the other hand, Sauwebrey equation is not valid for these cases, because it applies for thin and rigid films where energy losses are lower. Energy dissipation, D (or its inverse, the Q factor) was evaluated considering DMAa content in each HM, as exemplified in Figure 4.

Figure 4a shows a decrease in the nominal frequency *f*_0_ = 5 MHz induced to a 5 MHz quartz crystal microbalance (QCM) coated with the PHEMA HM. Within the time range from zero to 2400 ns, the oscillation is generated by a 5 MHz sinusoidal signal using a Tektronix model AFG 3022 model/arbitrary function generator. At 2400 ns, the signal was interrupted in order to verify how much time the signal maintained until the PHEMA-coated QCM was extinguished. Two changes in the oscillation were registered; a change in frequency from 5 MHz to 4971 MHz was observed, due to an increase in the mass of HM deposited, and a decrease in amplitude was recorded. Figure 4b shows the detail of this decrease in amplitude, which was adjusted to an enveloping function, similar to Equation (1).
(1)$$A\left(t\right)=A_{0}e^{\frac{t}{\tau }}sin\left(2ft+\theta \right)$$
where *A* is amplitude, *f* is the frequency and is the phase angle, and *τ* is the decay time constant of the envelope. The decay constant *τ* can be calculated from Equation (1) and is related to the dissipation factor *D* by Equation (2).
(2)$$\frac{1}{Q}=D=\frac{E_{Dissipated}}{2\Pi E_{Stored}}=\frac{1}{2\Pi f\tau }$$

The *D*-factor, from Equation (2), is related to the viscoelastic properties of the interface between the crystal QCM and each HM, and *τ* depends on the smoothness or rigidity of the material covering the QCM surface [52].

The estimated value of *τ* for PHEMA HM was 38.6 μs (Figure 4b), while for QCM coated with HM containing 40 mol% DMAa (Figure 4c) and for 100 mol% DMAa (Figure 4d), the values were 23.5 μs and 1.75 μs, respectively. According to Equation (2), lower *τ* is related to higher *D*-factor, thus that membrane with 100 mol% DMAa presented the highest dissipation factor, with a *τ* equal to 1.78 μs (Figure 4d and Figure 5a).

Studying the damping of the oscillation amplitude of the QCM sensor yields information on the viscoelastic properties of the HM coating the gold surface; as well as energy losses due to friction within the HM, which could be internal frictions between polymer chains of the oscillator itself, or due to the surrounding environment constituted by the components of the sample, such as water, analyte, and impurities. This friction causes the oscillatory energy to dissipate in the form of heat, then the HM in contact with the gold surface of the QCM will induce energy losses, which can be studied using the model of a harmonic oscillator that dissipates its energy from according to the following relationship. Therefore, the *τ* value for all membranes was determined in the same way, as shown in Figure 4a–d, and it was plotted with respect to the DMAa content of each membrane—the results are shown in Figure 5a. The DMAa content influences the viscoelastic properties of the HMs, which presented *τ* values from 38.6 μs to 1.75 μs, depending on the DMAa content.

On the other hand, the frequency variation from the nominal frequency obtained experimentally (as already detailed) to each HM is presented in Figure 5b. In none of the cases was frequency variation (Δ*f*) greater than 0.8%. Additionally, the density of each HM deposited on their corresponding QCM is shown in Figure 5c. As detailed in the characterization and techniques section, all the deposited membranes had an area of 1 cm^2^, and the deposited mass of each HM was determined experimentally by gravimetry. Hence, the density y is numerically equal to the deposited mass of each membrane. According to the equation proposed by Sauerbrey, there is a linear relationship between the resonance frequency of an oscillating quartz crystal and the mass changes on its gold surface, as shown in Equation (3).
(3)$$\Delta m=\frac{-C\times \Delta f}{n}$$
where Δ*m* is the mass variation over the gold surface of QCM; Δ*f* is the resonant frequency change; *C* is the mass sensitivity constant, which is related to the properties of the quartz and the membrane that covers the gold surface of the QCM, for a 5 MHz crystal, *C* is equal to 17.7 ng/(cm^2^ × z); and finally, n is the parameter related to the harmonic number, which can only be an odd number.

In all cases, the first harmonic was used, hence *n* = 1 for all hydrogel membranes (HMs); Δ*f* and Δ*m* values for each HM could be obtained from Figure 5b,c respectively. Therefore, the constant “*C*” has been determined for each HM, and the values are shown in Figure 5d. It can be seen that using the Sauerbrey equation was not possible to correctly calculate the value of the constant “*C*” for the membranes with 80 and 100 mol% DMAa. On the other hand, a better result was achieved for the membranes with 0 mol%, 20 mol%, and 40 mol% DMAa content. This is explained because the linear frequency-mass relationship described by Equation (3) is based on the behavior of a pure quartz crystal, of thickness h, and mass m, which will have a nominal resonance frequency, *f*_0_. If a layer of material similar enough to pure quartz is deposited on its surface, it can be approximated with a thicker crystal. The model that Sauerbrey proposed assumes that the deposited layer can be approximated to be part of the oscillating crystal itself, considering a new thickness (h + Δh). However, this assumption is only valid for layers deposited on the quartz crystal that are thin, rigid and firmly attached to the surface of the crystal. On the contrary, when the layer deposited on the quartz crystal is soft, thick or not coupled to the surface, Equation (3) fails.

### 2.6. Proof of Concept to QCM-Sensors

Considering the previous results from the evaluation of the QCM-sensors coated and sorption study, the membrane-coated QCM sensor with 60 mol% DMAa content was selected to conduct the performance test as the cation sensor used in the sorption study.

Therefore, to investigate for this test, the QCM sensor was left in contact with dH_2_O the previous night to reach equilibrium swelling (maximum water adsorption). Then, it was verified that the frequency of the QCM sensor with the swelled membrane was stable over time. Figure 6a shows the variations in the resonance frequency, from the nominal frequency of the uncoated QCM (5 MHz), of the coated QCM with the dried membrane of the copolymer with 60 mol% DMAa content (4.96 MHz), and of the coated QCM after 24 h under immersion (4.81 MHz). Later, the QCM with the membrane at the maximum swelling was placed in the acrylic container, an aliquot of 40 ppm Cd (II) solution was added to record the change in resonant frequency (Figure 6b). This figure shows that the QCM sensor had a stable frequency around 4.81 MHz with its membrane at maximum swelling, which was consistent with what was observed in Figure 6a for the same QCM sensor. Approximately 40 s later, an aliquot of the Cd (II) solution was added with a concentration of 40 ppm, and ca. 20 s later, the sensor stabilized at a new resonance frequency, with a frequency shift (Δ*f*) of 169.6 Hz. This procedure was repeated with each of six cations with each of the different concentrations from 10 ppm to 200 ppm, and the frequency shifts obtained are shown in Figure 6c.

In all cases, there was an increase in the values of −Δ*f*, due to an increase in the concentration of the adsorbate (cations) at concentrations less than 120 ppm (Figure 6c), but no significant frequency shift was recorded at concentrations equal to or greater than 120 ppm. This is consistent with the adsorption isotherm shown in Appendix A, where it was observed that the adsorption profiles presented an asymptote behavior between 120 ppm and 180 ppm; which means that the membranes become saturated with a monolayer of ions around these initial concentrations.

A quasi-linear behavior was found for both adsorbates for initial concentrations from 10 ppm to 120 ppm, as shown in Figure 6d; i.e., within the concentration range of 10 to 120 ppm, all analytes or ions were quantitatively detected individually using the HM(40 mol% DMAa)-coated QCM sensor.

## 3. Conclusions

In this study, six hydrogel membranes (HMs) using HEMA and DMAa as comonomers, were synthesized at different molar ratios of the comonomers, and the effects of the DMAa content on the swelling and sorption properties were studied. Additionally, through in situ polymerization on the gold surface of the quartz crystal microbalances (QCM), a QCM sensor for each of the six HMs was prepared to study the effect of DMAa on the viscoelastic properties of the membranes. The results obtained between the decay time constant (*τ*) and the %mol DMAa content showed that the DMAa present in the materials produce softer membranes and the Sauerbrey equation was not valid for concentrations higher than 60%mol of DMAa content.

The membrane-coated QCM with 60%mol DMAa content was selected to perform a proof of concept, and to explore and illustrate how a QCM-type sensor can be designed and manufactured using these membranes. Six aqueous solutions of different cations (10–200 ppm) were prepared to evaluate copolymers adsorption capacity. Results showed that the cations tested were susceptible to being adsorbed into the membranes, but with different physicochemical parameters of adsorption, such as Qe, Ce, Qe/Co and equilibrium time. Therefore, the HM(60%mol DMAa)-coated QCM sensor is expected to be sensitive to a variety of cations. The results of resonance frequency shift and variation of cation concentration showed a good linear correlation for concentrations lower than 120 ppm (for all the cations studied). The membrane (with 60 mol% DMAa content) has a differentiated response for each cation, then it can be used as a membrane to encapsulate recognition elements and even be used by itself as a recognition element; because it is possible to determine each cation by the slope of the linear fit. Therefore, the hypothesis is correct for the membrane used for the proof of concept.

The results presented in this report are promising with good prospects for the development of future biosensing devices. Therefore, it will be interesting to delve into the characterization and study of membranes based on HEMA copolymerized with DMAa, or even other more complex amines, as well as propose future studies to immobilize some recognition probes inside the HEMA-co-DMAa membranes.

## 4. Materials and Methods

### 4.1. Materials

Monomers: 2-hydroxyethyl methacrylate (HEMA; <97%, CAS 868-77-9) and N,N-dimethylacrylamide (DMAa; <99%, CAS 2680-037); free radical initiator: α,α′-azoisobutyronitrile (AIBN; <99%, CAS 78-67-1); reagents products to be removed: lead (II) nitrate (Pb(II); <99%, CAS 10099-74-8), Copper(II) nitrate trihydrate (Cu(II); <99%, CAS 10031-43-3), Cobalt(II) nitrate hexahydrate (Co(II); <98%, CAS 10026-22-9), Cadmium(II) acetate (Cd(II); <99%, CAS 543-90-8), Chromium(III) nitrate nonahydrate (Cr(III); 99%, CAS 7789-02-8), Iron(III) nitrate nonahydrate (Fe(III); <99.9%, CAS 7782-61-8), citric acid (C6H8O7; >99.5%, CAS 77-92-9), sodium hydroxide concentrate (NaOH; in water 0.1N, CAS 1310-73-2) and trans-1,2-Diaminocyclohexane-N,N,N′,N′-tetraacetic acid monohydrate (DACT; <99%, CAS 125572-95-4); and solvents such as methanol (MeOH; 99.9%, CAS 67-56-1) were supplied by Merck (Sigma-Aldrich, Toluca, Mexico or St. Louis, MO, USA) and were used without further purification, except for monomers and radical initiator. Inhibitors from HEMA and DMAa were removed by adsorption (Inhibitor remover, Sigma-Aldrich 306312 product number), and AIBN was recrystallized from cold methanol and dried at room temperature under vacuum prior to use. Finally, all experiments were carried out using deionized water (dH_2_O).

### 4.2. Hydrogel Membranes Synthesis

Copolymer poly(HEMA-co-DMAa) hydrogels were synthesized by a free-radical in bulk copolymerization from six mixtures, each one with different monomeric ratios (varying the DMAa monomer ratio from 0% to 100% in 20% increments, to obtain six HEMA-based hydrogel membranes). Samples were labelled copolymer1 (C1) to copolymer6 (C6) (HEMA/DMAa ratio in %mol: C1 1000/0; C2 80/20; C3 60/40; C4 40/60; C5 20/80; and C6 0/100), thus C1 and C6 were the homopolymers of PHEMA and PDMAa, respectively. AIBN (Azobisisobutyronitrile) was used as an initiator at 55 °C, without stirring, and under a nitrogen atmosphere. Total volume in each monomeric mix was ca. 1.5 mL, which was added to a 5 mL glass vial (inner diameter = 12 mm), with a screw cap and a hole with PTFE/silicone septum. After 15 min of ultrasound stirring at a controlled temperature of 15 °C, a solution of AIBN in methanol (ca. 0.1 mL; 0.055 g × mL^−1^) was added and degassed using N_2_ by standard Schlenk technique. Polymerization was carried out at 55 °C for ca. 40 min. The obtained material was washed on a daily basis with dH_2_O. The resulting hydrogel was freeze-dried and cut to obtain HMs with uniform dimensions (diameter 11.5 mm and thickness of 3 mm) for further characterization. Figure 7a,b shows a schematic of the hydrogel membranes preparation.

### 4.3. Characterization and Techniques

HMs were chemically characterized by attenuated total reflectance–Fourier transform infrared (ATR-FTIR) spectroscopy, in the range of 4000–400 cm^−1^, with a resolution of 4 cm^−1^ and accumulation of 24 scans.

Since the objective was to evaluate the general physicochemical swelling and sorption properties depending on the DMAa content in each membrane, these properties were evaluated by immersion in deionized water at a temperature of 25 °C using a temperature-controlled water bath. Ion solutions of lead(II), copper(II), cobalt(II), cadmium(II), chromium(III), and iron(III) were used in sorption studies at a controlled temperature and pH of the medium. For more details, see the swelling measurements and sorption studies sections within the Supplementary Information.

Subsequently, quartz crystal microbalances (QCM) coated with membranes of each of the synthesized copolymers were prepared; six coated QCMs were prepared in total. Meanwhile, the gold surface of each one quartz crystal microbalance (QCM, AT-Cut 25 °C, Maxtek^®^, Cr/Au, 5 MHz Etched, manufactured by Inficon Ltd., East Syracuse, NY USA) was cleaned with piranha solution (30% H_2_O_2_:H_2_SO_4_, 1:3, *v*/*v*), followed by sequential deionized water and ethyl alcohol washings, and was vacuum oven-dried (200 mmHg, 40 °C) for 3 h.

Then, the QCM-sensor surface cleaned was immersed in allyl thiol/ethanol solution (3 mM) for 15 h to attach a self-assembled monolayer of thiol groups on the gold surface of the QCM crystal electrode. Subsequently, the QCM functionalized was washed to remove the reactant excess and dried under N_2_. To assemble each membrane on QCM, the thiol-modified QCM was placed in the bottom of a sealed glass container, reminiscent of a contact lens case. Then, 10 μL (ca.) of a degassed emulsion of HEMA, DMAa, and AIBN solution (prepared as mentioned above for each HMs) was dispensed over the surface of the thiol-modified gold electrode. The polymerization was performed inside a temperature-controlled oven, in the same conditions as the hydrogel. The resultant QCMs coated were cleaned from remnant chemicals with deionized water. Studies of each membrane viscoelastic properties were performed by placing each coated QCM on an acrylic chamber (own manufacture) to induce a 5 MHz sinusoidal signal by a Tektronix-brand arbitrary/function generator (model AFG 3022; frequency range: 1 mHz–25 MHz; amplitude: 10 mVp-p to 10 Vp-p). This signal had the following parameters: precision (stability) of ± 1 ppm at 25 °C, as well as 1 µHz resolution and a 1.4 VP-P amplitude. The driving voltage was then turned off and its decay over the crystal was recorded as a function of time. An MSO 4034 mixed-signal digital oscilloscope with bandwidth up to 350 MHz, 2.5 GS/s, and time base range from 1 ns to 1000 s was used for the analyses.

Once the six coated QCMs were characterized (under dry coating membrane conditions), the QCM with 60%mol DMAa content was selected to conduct the proof of concept, as described below. The HM(60 mol% DMAa)-coated QCM was immersed in distilled water the previous day (24 h immersion) and the resonance frequency of the swollen membrane was recorded (fsw) and compared against the nominal resonance frequency (*f*_0_) of the same uncoated QCM. Subsequently, the resonance frequency was recorded when filling (ca. with 5 mL) the acrylic chamber with the coated QCM (with the membrane swollen) inside, using progressive concentrations from 10 ppm to 200 ppm, of ionic solutions of lead (II), copper (II), cobalt (II), cadmium (II), chromium (III) and iron (III), at the same temperature and pH conditions as sorption studies.

## Figures and Tables

**Figure 1 gels-07-00151-f001:**
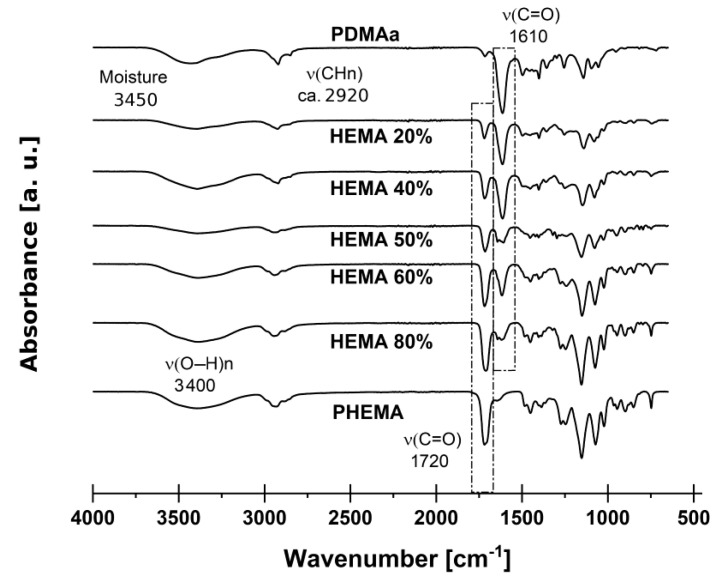
ATR-FTIR spectra of poly(HEMA-co-DMAa) copolymers.

**Figure 2 gels-07-00151-f002:**
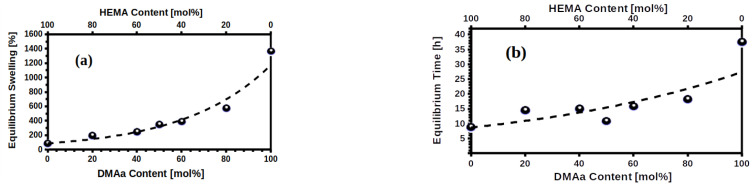
(**a**) Maximum swelling (equilibrium) and (**b**) Time required to reach maximum swelling; as function of %mol DMAa content, at 25 °C and pH 7.

**Figure 3 gels-07-00151-f003:**
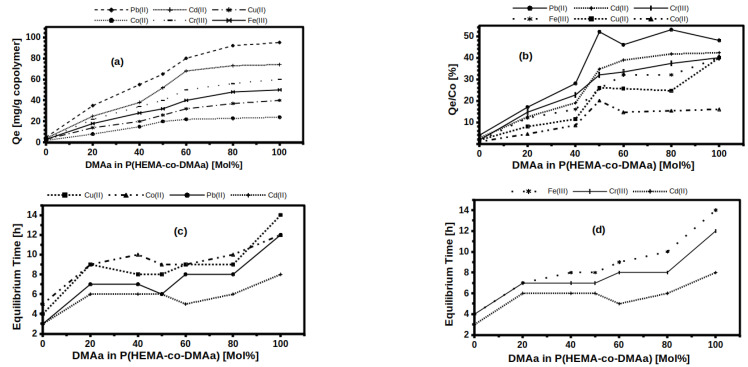
(**a**) Qe as function of %mol DMAa content; (**b**) Qe/Co as function of %mol DMAa content; (**c**) Equilibrium time as function of %mol DMAa content for ions 2^+^; (**d**) Equilibrium time as function of %mol DMAa content for ions 3^+^.

**Figure 4 gels-07-00151-f004:**
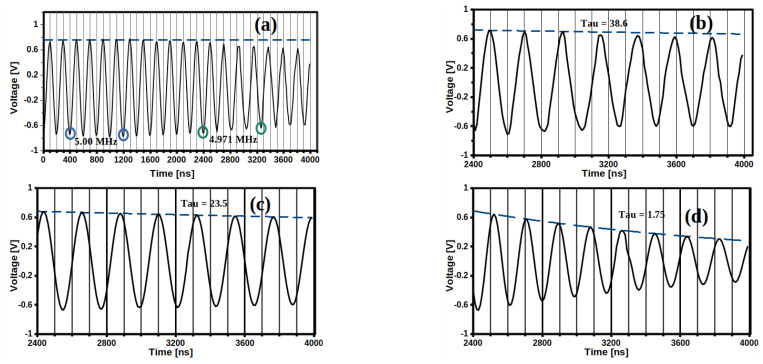
Voltage oscillation frequency as a function of time: (**a**) membrane without DMAa [P(HEMA)]; (**b**) Detailed region of the 2400 to 4000 ns range of membrane without DMAa; (**c**) membrane with 40%mol DMAa; (**d**) membrane with 100%mol DMAa [P(DMAa)].

**Figure 5 gels-07-00151-f005:**
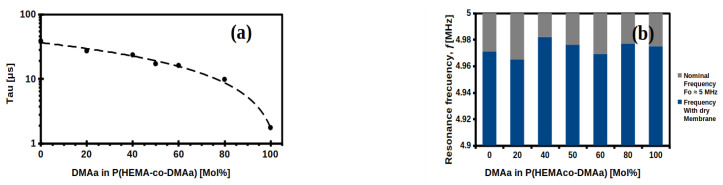
(**a**) Tau constant as a function of DMAa content in Hydrogel Membranes (HM) at 25 °C; (**b**) Resonance frequency of the HMs-coated QCM sensor at different %mol DMAa content; (**c**) Density for each once HMs-coated QCM sensor; (**d**) Mass sensitivity constant [C] of each once HMs-coated QCM sensor, calculated from Figure 5b.

**Figure 6 gels-07-00151-f006:**
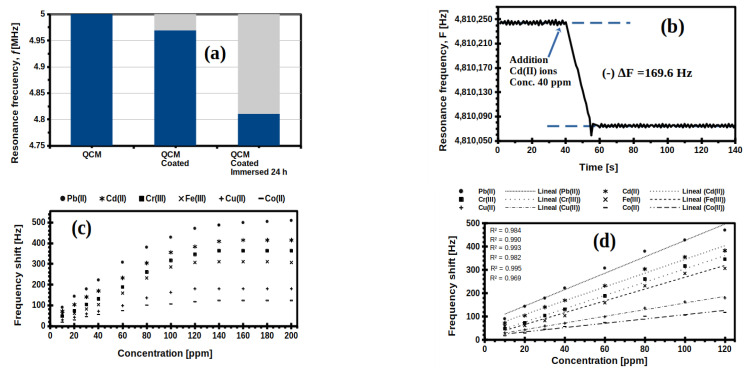
(**a**) Resonance frequency shift of the HM(60 mol% DMAa)-coated QCM; (**b**) Time course of resonance frequency, *f*, of the HM(60 mol% DMAa)-coated QCM sensor exposed to dH2O, followed by Cd(II) ions solution added at 40 ppm and at 25 °C; (**c**) Resonance frequency shift, −Δ*f*, of the HM(60 mol% DMAa)-coated QCM sensor at different concentrations, from 10 ppm to 200 ppm of each one ions and at 25 °C; (**d**) Linear fits of data from Figure 6c within the concentration range of 10 ppm to 120 ppm.

**Figure 7 gels-07-00151-f007:**
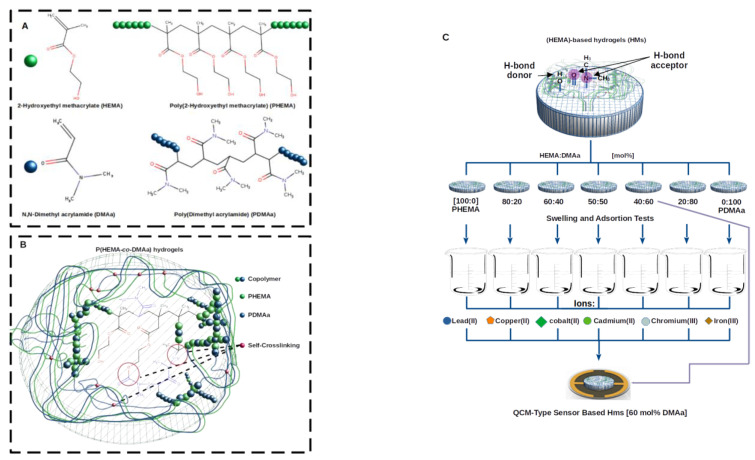
Schematic representation for the; (**A**) monomers HEMA and DMAa, and their respective polymers; (**B**) P(HEMA-co-DMAa) hydrogel network structure representation; and (**C**) swelling, adsorption, and QCM sensor test.

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
