# Peer review of "Poly(2-Hydroxyethyl methacrylate-co-N,N-dimethylacrylamide)-Coated Quartz Crystal Microbalance Sensor: Membrane Characterization and Proof of Concept"

_gels, 2021, doi:10.3390/gels7040151_

Round 1

Reviewer 1 Report

In this study, synthesized a series of Poly(2-hydroxyethyl methacrylate-co-N,N-dimethylacrylamide) hydrogel membranes at different molar ratios to  gain insight into the effects of these constituting elements (monomers) on the overall swelling and sorption properties, as well as in their transport phenomena. Finally, a proof of concept was established to explore and illustrate the way in which a working quartz crystal microbalance-type (QCM). The membrane-coated QCM with 60 %mol DMAa content was selected to perform proof of concept and to explore and illustrate how a QCM-type sensor can be designed and manufactured using these membranes. The results presented in this report are promising with good prospects for the development of future biosensing devices. Thus, I recommend it for publication in Gel after a minor revision. Some issues need to be addressed:

Q1. In the title of the article QCM should write full form ¨quartz crystal microbalances¨

Q1. In Supplementary, sorption studies section equation 2 and 3 need to be fixed

Author Response

Dear Reviewer, I appreciate your contribution and below I comment on each suggestion

Q1. In the title of the article QCM should write full form "quartz crystal microbalances"
Answer 1: Article title has been modified

Q2. In Supplementary, sorption studies section equation 2 and 3 need to be fixed
Answer 1: equation 2 and 3 were fixed

Reviewer 2 Report

Hernandez-Martinez reported a p(HEMA-co-DMAa) hydrogel-coated QCM sensor for metal cation sensing. The author synthesized a series of copolymer hydrogels varying the ratios of HEMA and DMAa, and compared the swelling and sorption properties using various metal cations. In addition, the hydrogels were coated on the gold surface of quartz crystals, and the oscillation property and dissipation factor were analyzed thoroughly. Lastly, the QCM responses for sorptions of metal cations were tested and analyzed. The characterizations and analyses were conducted reasonably well and the work would be helpful for the researchers in the related field. Thus this manuscript seems worthy of publication in the journal Gels after addressing the following issues.

  1. The structure of the copolymer hydrogels should be modified. Especially, the chemical structure of DMAa in Figure 7A seems not correct. In addition, the crosslinked or network structure of the polymer gel should be presented. As the author mentioned, DMAa is the self-crosslinking acrylamide, and representing the crosslinked structure will be helpful for the readers to understand.
  2. In page 2 (lines 71-73), the author stated that “HMs applications for chemical sensor development and for human diagnosis tests from…(HEMA)-based hydrogels have not been widely studied,” which can be misunderstood as the first report on this material. The author may need to modify the sentence. Similarly, the statement “this is the first report on HMs form P(DMAa-co-HEMA) on page 3 (lines 106-107) seems exaggerated since several reports on the hydrogels already existed.
  3. The swelling behavior seems well conducted (Figure 2 and Fig. S1). However, why did the hydrogels with higher portions of DMAa show the higher swellings? Since DMAa is the so-called “crosslinking reagent”, one can imagine the more DMAa, the more crosslinking points, resulting in the less swelling. The author should comment on this issue, and should be reflected in the hydrogel structure. In fact, the author mentioned the increased hydrophilicity due to DMAa compared to HEMA. However, based on the chemical structures, both monomers (DMAa and HEMA) are hydrophilic and it is hard to differentiate between them.
  4. In page 5 (line 176), what is “RMN”? The acronyms should be expressed as a full name first.
  5. In page 7 (line 229), the authors described the interaction between cations and amide groups as “Van Der Waals forces”, which seems not correct. The van der Waals or dispersion forces occur normally between non-ionic, neutral (but instantly polarizable) molecules.
  6. The author explained the different sorption properties of metal cations in terms of the ionic radius. First of all, the radius values need proper reference(s). In addition, the strength of metal-ligand (i.e., amide) interactions should be supported by proper reference(s).
  7. Figure 6c represented the frequency shifts according to the metal cations, which should be related the amount of absorbed ions. Thus the authors may discuss these results with the sorption results in Figure 3.
  8. It would be very helpful if the limit of detections (LODs) for each metal ion are presented.
  9. Several typos are found. The author needs to check out the manuscript thoroughly.

Author Response

Reviewer 2

Hernandez-Martinez reported a p(HEMA-co-DMAa) hydrogel-coated QCM sensor for metal cation sensing. The author synthesized a series of copolymer hydrogels varying the ratios of HEMA and DMAa, and compared the swelling and sorption properties using various metal cations. In addition, the hydrogels were coated on the gold surface of quartz crystals, and the oscillation property and dissipation factor were analyzed thoroughly. Lastly, the QCM responses for sorptions of metal cations were tested and analyzed. The characterizations and analyses were conducted reasonably well and the work would be helpful for the researchers in the related field. Thus this manuscript seems worthy of publication in the journal Gels after addressing the following issues.

Thank you for your kind comments, I approached to them as follows;

Q1. The structure of the copolymer hydrogels should be modified. Especially, the chemical structure of DMAa in Figure 7A seems not correct. In addition, the crosslinked or network structure of the polymer gel should be presented. As the author mentioned, DMAa is the self-crosslinking acrylamide, and representing the crosslinked structure will be helpful for the readers to understand.

Answer 1: Figure 7 was replaced with a new original figure that shows the corrected chemical structures and the cross-linked structure of the polymer gel.

Q2. In page 2 (lines 71-73), the author stated that “HMs applications for chemical sensor development and for human diagnosis tests from…(HEMA)-based hydrogels have not been widely studied,” which can be misunderstood as the first report on this material. The author may need to modify the sentence. Similarly, the statement “this is the first report on HMs form P(DMAa-co-HEMA) on page 3 (lines 106-107) seems exaggerated since several reports on the hydrogels already existed.

Answer 2: Lines 71-73 were deleted and the correct idea was integrated in the rest of the text in the introduction. Sentences in lines 106-107 were modified.

Q3. The swelling behavior seems well conducted (Figure 2 and Fig. S1). However, why did the hydrogels with higher portions of DMAa show the higher swellings? Since DMAa is the so-called “crosslinking reagent”, one can imagine the more DMAa, the more crosslinking points, resulting in the less swelling. The author should comment on this issue, and should be reflected in the hydrogel structure. In fact, the author mentioned the increased hydrophilicity due to DMAa compared to HEMA. However, based on the chemical structures, both monomers (DMAa and HEMA) are hydrophilic and it is hard to differentiate between them.

Answer 3: An explanation of why hydrogels with higher portions of DMAa show higher swellings is because self-crosslinking can occur in more than one way in the copolymers synthesis. Both PDMAa and PHEMA presented self-crosslinking separately. There are reports on self-crosslinking of both DMAa [10.1016/j.ijpharm.2013.07.073; 10.3390/polym12061401; 10.1201/9780203507438.ch3]  and HEMA [Muter et al., Int. J. Res. Stud. Biosci, 3(2015); 10.1021/acs.biomac.9b01196]. The self-crosslinking of DMAa occurs mainly between its methylene and carbonyl groups. While HEMA can present self-crosslinking between its hydroxyl and carbonyl groups. On the other hand, hydrophilicity is related to functional groups and despite the fact that both DMAa and HEMA are hydrophilic, they have different functional groups, both provide carbonyl groups, but DMAa provides amino groups, while HEMA provides hydroxyl groups. Our experimental data confirm that PDMAa is more hydrophilic than PHEMA, which is consistent with several previous reports and therefore we affirm that higher DMAa contents increase hydrophilicity due to the combination of hydrophilic functional groups. Figure 7 has been modified to reflect this in the hydrogel structure representation.

Q4. In page 5 (line 176), what is “RMN”? The acronyms should be expressed as a full name first.

In page 7 (line 229), the authors described the interaction between cations and amide groups as “Van Der Waals forces”, which seems not correct. The van der Waals or dispersion forces occur normally between non-ionic, neutral (but instantly polarizable) molecules.

Answer 4: Acronym in line 176 was corrected with the full name of the technique, since is the only mention of it in the manuscript. The phrase "Van Der Waals forces" has been changed to "intermolecular attractive forces".

Q5. The author explained the different sorption properties of metal cations in terms of the ionic radius. First of all, the radius values need proper reference(s). In addition, the strength of metal-ligand (i.e., amide) interactions should be supported by proper reference(s).

Answer 5: The following references were added:

(i) R. D. Shannon, (1976). Revised effective ionic radii and systematic studies of interatomic distances in halides and chalcogenides, Acta Cryst., A32, 761. doi 10.1107/S0567739476001551

(i) M. Birkholz and R. Rudert, (2008). Interatomic distances in pyrite-structure disulfides – a case for ellipsoidal modeling of sulfur ions., physica status solidi, 245(9) 1858-1864.. doi 10.1002/pssb.200879532

as references [47,48] in the corresponding section (2.3. sorption study).

(iii) David J.BarnesRoss L.ChapmanFrederick S.StephensRobert S.Vagg, (1981). Studies on the metal-amide bond. VII. Metal complexes of the flexible N4 ligand N,N′-bis(2′-pyridinecarboxamide)1,2-ethane. Inorganica Chimica Acta, 51, 155-162. doi 10.1016/S0020-1693(00)88333-4.

(iv) Chapman Ross.L.,Stephens F.S.,Vagg R.S. (1980). Studies on the metalamide bond. II. The crystal structure of the deprotonated copper(II) complex of N,N′-bis-(2′-pyridinecarboxamide)-1,2-benzene,  43 C, 29 - 33. doi 10.1016/S0020-1693(00)90500-0

(v) Cary R. Stennett, Clifton L. Wagner, James C. Fettinger, Petra Vasko, and Philip P. Power, (2021). Reductions of M{N(SiMe3)2}3 (M = V, Cr, Fe): Terminal and Bridging Low-Valent First-Row Transition Metal Hydrido Complexes and “Metallo-Transamination”, Inorganic Chemistry 60 (15), 11401-11411. doi 10.1021/acs.inorgchem.1c01399.

as references [49-52] in the corresponding section (2.3. sorption study).

Q6. Figure 6c represented the frequency shifts according to the metal cations, which should be related the amount of absorbed ions. Thus the authors may discuss these results with the sorption results in Figure 3.

Answer 6: Figure 3 shows the physicochemical equilibrium state, but Figure S3 (supplementary material) shows the equilibrium and transition states; therefore the latter was selected to address your kind comment. In consequence, a discussion of results between the figure 6c and the adsorption isotherms of figure S3 was added.

Q7. It would be very helpful if the limit of detections (LODs) for each metal ion are presented.

Answer 7: I agree that the limit of detection is a key data in chemical sensing. However, in this initial report the objective is to demonstrate the feasibility of manufacturing sensors using the presented copolymer. The results do not allow us to conclude about LODs because we only report within the 10-200 ppm range. But we are working on detection tests at ultra-low concentrations, less than 10 ppm because now we know that the QCM used for the sensors, has sensitivities and LOD in the order of ppb. Obtaining the calibration curve represents a great challenge for my team, because currently we only have UV-vis spectroscopy available, then we are looking for collaborations to access other techniques of greater sensitivity such as atomic absorption spectroscopy.

Q8. Several typos are found. The author needs to check out the manuscript thoroughly.

Answer 8: The manuscript was thoroughly revised and corrected.

Reviewer 3 Report

In the present work, the author synthesizes a series of P(DMAa-co- HEMA) hydrogel membranes (HMs), at different molar ratios of the co-monomers and then investigated the effects of monomers on the swelling, sorption, and transport behavior of the HMs. Moreover, the author also attempted to establish a proof of concept to design and fabricate the P(DMAa-co- HEMA)-HMs -based quartz crystal microbalance-type (QCM) sensor.

Overall in the study, the author prepared HMs and tested them as QCM -sensors. This type of study can be considered as an initial attempt to establish a biosensor that requires extensive further investigations.

 As Gels Journal encourages authors to publish initial findings, I strongly recommend the manuscript for possible consideration of publication with minor modifications.

  1. The number of words needs to be reduced in the introduction part as its looks like a review article and the quality of presentation in each section should be refocused related to the objective of the work.
  2. The overall quality of the figure and figure legend needs to be improved.
  3. The schematic sketch should be further improved

Author Response

Reviewer 3

In the present work, the author synthesizes a series of P(DMAa-co- HEMA) hydrogel membranes (HMs), at different molar ratios of the co-monomers and then investigated the effects of monomers on the swelling, sorption, and transport behavior of the HMs. Moreover, the author also attempted to establish a proof of concept to design and fabricate the P(DMAa-co- HEMA)-HMs -based quartz crystal microbalance-type (QCM) sensor.

Overall in the study, the author prepared HMs and tested them as QCM -sensors. This type of study can be considered as an initial attempt to establish a biosensor that requires extensive further investigations.

 As Gels Journal encourages authors to publish initial findings, I strongly recommend the manuscript for possible consideration of publication with minor modifications.

Q1. The number of words needs to be reduced in the introduction part as its looks like a review article and the quality of presentation in each section should be refocused related to the objective of the work.

Answer 1. The introduction was substantially modified, as well as some sections; changes are highlighted in the document.

Q2. The overall quality of the figure and figure legend needs to be improved.

Answer 2. Figures have been uploaded at 500 dpi quality, captions have been revised. 

Q3. The schematic sketch should be further improved

Answer 3. The schematic sketch has been modified and improved to be helpful for the readers.